# Imidacloprid decreases energy production in the hemolymph and fat body of western honeybees even though, in sublethal doses, it increased the values of six of the nine compounds in the respiratory and citric cycle

Jerzy Paleolog[1]*, Jerzy Wilde[2], Marek Gancarz[3,4,5], Aneta Strachecka[1]

1 Department of Invertebrate Ecophysiology and Experimental Biology, University of Life Sciences in Lublin, Lublin, Poland, 2 Department of Poultry Science and Apiculture, Faculty of Animal Bioengineering, Warmia and Mazury University in Olsztyn, Olsztyn, Poland, 3 Faculty of Production and Power Engineering, University of Agriculture in Kraków, Kraków, Poland, 4 Institute of Agrophysics, Polish Academy of Sciences, Lublin, Poland, 5 Center of Innovation and Research on Healthy and Safe Food, University of Agriculture in Kraków, Kraków, Poland

* jerzy.paleolog@up.lublin.pl

## Abstract

### Background

Neonicotinoids, including imidacloprid (IM), cause harm to *Apis mellifera* in a number of ways, among others by impairing body maintenance, resistance and immunity. Energy resources are important to preventing this, as we hypothesized, not only in the hemolymph but particularly in the fat body, the insufficiently investigated, as yet, equivalent of the mammalian liver and pancreas. Both suppression and horme-sis (diaphasic stressor response) of energy supply was reported in the energy-dependent traits of bees exposed to sublethal doses of imidacloprid. Therefore, our goal was to answer which of these two phenomena occurs in the hemolymph/fat body and at what doses of imidacloprid.

### Methods

concentrations/activities of respiratory/citric cycle compounds (acetyl-CoA, IDH-2, AKG, succinate, fumarate, NADH2, COX, UQRC, and ATP) were compared in the hemolymph and fat bodies of nurse workerbees sampled from honeybee colonies fed with diets containing 200 ppb (IM-200), 5 ppb (IM-5; sublethal), and 0 ppb of IM in a field experiment.

### Results

the assayed compounds had higher values in the fat body than in the hemolymph, whereas their variability was higher in the hemolymph. The pattern of response to IM

**Data availability statement:** The raw data are held in a public repository "RepOD" and are avaliable by DOI: https://doi.org/10.18150/MMKUSE.

**Funding:** AS; The founding no. is: LKE.SUBB. WLE.22.058 by Ministry of National Education of the Republic of Poland, via University of Life Science in Lublin, Poland, as well as, that the funders had no role in study design, data collection and analysis, decision to publish, or preparation of the manuscript.

**Competing interests:** The authors have declared that no competing interests exist.

was the same in both tissues, but markedly differed between IM-200 and IM-5. The concentrations of the strongly correlated NADH2, ATP and acetyl-CoA decreased both in IM-200 and IM-5, whereas the levels of the other compounds decreased in IM-200 but increased in IM-5.

## Conclusions and significance

decreased ATP and acetyl-CoA levels both in IM-5 and IM-200 show that the pesticide impairs the hemolymph and fat-body energy metabolism in spite of hormesis in six of the nine respiratory and citric cycle compounds even in low, residual doses. This finding better explains how residual doses of neonicotinoids may disturb the fat body functions, and therefore suppress the apian resistance, which expands our knowledge about honeybee colony losses.

---

## Introduction

Exposure to pesticides, and among them to neonicotinoids, is believed to be one of the main factors blamed for the depopulation or mortality of honeybee (*Apis mellifera* L.) colonies [1–3]. Due to the crucial role of these insects in crop pollination and plant biodiversity [4] the global concern for their survival and health also involves the pesticide threat [5,6]. Among pesticides, imidacloprid [1-(6-chloro-3-pyridylmethyl)-2-nitroimino-imidazolidine] is the most commonly used neonicotinoid insecticide all over the world, particularly in both Americas and Asia. Even though its use has been forbidden in the European Union, it is still applied there for many "special" reasons, including indoor applications, e.g. in greenhouses [7]. Moreover, some of these insecticides may accumulate in the bee wax [8]. As wax reuse is a common practice, this additional source of honeybee exposure to imidacloprid should be taken into consideration.

Similarly to the remaining neonicotinoids, imidacloprid leads to insect neurotoxicity and harms the non-target honeybees by deteriorating numerous performance and survival traits, such as the life-span, reproduction, health, learning, immunity, and resistance to xenobiotics in many ways [9–11]. Moreover, its residues are present across ecosystems, particularly rural ones [12,13]. Notably, sublethal, even residual, doses of imidacloprid may degrade the honeybee health status, as well [14,15]. The biochemical backgrounds of these adverse effects need, however, more in-depth explanation, not only to better understand the mechanisms of the bee colony decline but to better protect the colonies [16].

The energy production is crucial for the insect organism defense in adverse environments, where it is exposed to pathogens, xenobiotics or nutritional stressors. In other words, energy is involved in the so-called body-maintenance (or self-maintenance) system that is pivotal for the healthy functioning [17], survival, and more generally, for the successful evolution [18,19] of the social insects. Therefore, the expression of genes responsible for energy metabolism and those related to mitochondrial functions were studied not only in honeybees [7,20], but also, for example,

in bumblebees [21] exposed to neonicotinoid pesticides. On the other hand, the neonicotinoids were shown to extend the homing flight time, increase $CO_2$ production by the colony, reduce the standard metabolic rate, and cause bee forager exhaustion, since they are believed to suppress the energetic metabolism and energy allocation, probably by disturbing mitochondrial functions [20,22–27].

There is, however, a problem with the exposition of honeybees to sublethal, residual, field-relevant doses of imidacloprid (about 5 ppb), as different results of the exposition were obtained in different studies. This implies a yet incompletely explored complex response to this pesticide. On the one hand, unlike the higher doses, which have consistently proven to be highly detrimental, they were shown to stimulate foraging, arousal and bee activity, or profitably improve hive thermoregulation [23,28]. Clothianidin, a neonicotinoid pesticide, increased the flight efficiency of monarchs (*Danaus plexippus*), the migratory butterflies, as well, by stimulating their cognition and motor functions, but impaired painted lady (*Vanessa cardui*) flights [29]. On the other hand, Tosi et al. [5] reported that sublethal doses of neonicotinoids can reduce flight and foraging abilities and our previous publications revealed that an addition of 5 ppb of imidacloprid to the honeybee worker diet deteriorated many physiological processes, especially those related to the functioning of the antioxidant and proteolytic systems or electrolyte/bioelement metabolism [15]. Kim et al. [30] showed that 5 ppb and 20 ppb neonicotinoid additions to the honeybee food decreased the body weight of the foragers, but 100 ppb increased it. Such a response did not occur in the nurse bees. The forager flight performance, however, always decreased independently of the pesticide concentration. One way or another, it seems that the biochemical processes underlying the above-mentioned differing responses to imidacloprid, are not well-understood yet and too little is known about them. Consequently, more studies are necessary to answer whether body energy supply is decreased by sublethal doses of the pesticide or whether such doses can also increase the supply as a result of hormesis.

The second problem is that it was mostly brain transcriptome analyses that were applied in order to find out whether and how exposure to neonicotinoids may alter the honeybee energy metabolism. Similar treatment was undertaken in the case of acetylcholine, vitellogenin, the stress response, antioxidative defense, detoxification, and immunity (compare Christen et al. [20,24,25], Fent et al. [7], Martelli et al. [31]). Therefore, we used a different approach in this study. Instead of applying the brain transcriptomics, we holistically assayed the pivotal set of nine compounds involved in the energy metabolism (including those involved in the mitochondrial electron transport chains) in two metabolically important honeybee tissues, i.e. the hemolymph and fat body. We considered Acetyl Coenzyme A (acetyl-CoA), Isocitrate dehydrogenase (IDH-2), Alpha-Ketoglutarate (AKG), succinate, fumarate, nicotinamide adenine dinucleotide (NADH2), Cytochrome c Oxidase (COX), Cytochrome c reductase (UQCR), and Adenosine triphosphate (ATP). There is not much data available on the concentrations/activities of these compounds in bees, and, as mentioned above, previous studies mostly focused on gene expression. Hence, our research fills in the gap in the knowledge about the role of these compounds, essential for the Krebs cycle/respiratory chain, in bees exposed to imidacloprid. Finally, hemolymph is a tissue that has been quite often analyzed in bees, since as "a carrier" it contacts and supplies many apian tissues and participates in their metabolism. Therefore, it can be treated as a good reference for other tissues. On the other hand, information about the metabolism and functions of the apian fat body is fragmentary. However views on its role have been substantially changing over the last decade [32]. Consequently, we wanted to fill in this gap in the knowledge about the effects of imidacloprid on the most important processes of cellular respiration, energy metabolism pathways and total energy in the honeybee fat body in comparison with the hemolymph. We also tried to answer whether dose-independent suppression or rather hormesis could be considered in this case.

Many toxicological experiments on pesticides in honeybees were performed in cage or semi-cage experiments [14,22] that allowed to better monitor/control the external conditions. However, results obtained in fully functional colonies in the field could be different. Therefore, we decided to perform a field experiment in this study, controlling environmental factors to the maximum extent.

The aim of this study was to compare the concentrations/activities of the above compounds in workerbees captured from the colonies fed with a diet containing 200 ppb, 5 ppb or 0 ppb of imidacloprid. Both the hemolymph and fat bodies of the workerbees were assayed.

## Materials and methods

### Rearing of workerbees exposed or unexposed to imidacloprid

The synthetic colonies of similar strength, as well as worker and brood structure, were set up in northeastern Poland (19.53 E, 53.50 N) in June 2024. Each of these colonies fully populated a one-box hive with ten 360 mm x 260 mm frames and was headed by an egg laying one-year-old purebred queen. All the queens belonged to the same *Apis mellifera carnica* commercial stock and were all obtained from the same mother-queen. In the first ten days of June, after removing the hive food-stores, the colonies were given sugar/water syrup (5:3 w/w) containing 0 ppb, 5 ppb or 200 ppb of imidacloprid (Bayer Health Care AG, Leverkusen, Germany) to obtain, respectively, three feeding groups: IM-0 (the control group) or IM-5 and IM-200 (two experimental groups). There were three colonies in each group. The syrup was added every 5 days for six weeks. The unused leftovers were removed at the same time interval. The diets in IM-0, IM-5 and IM-200 were replenished with bee-food (API-Fortune HF 1575, Bollène, France) containing accordingly 0 ppb, 5 ppb or 200 ppb of imidacloprid. The imidacloprid concentration of 5 ppb is considered sublethal and similar to that chronically observed in the field (residuals). This concentration sometimes turn out to be adverse for honeybee colonies either impairing the physiology of an individual bee or weakening entire colonies – particularly in a long-term perspective [15,16,28]. Notably, its negative symptoms may be hidden. On the other hand, the concentration of 200 ppb is considered harmful or severely harmful and even lethal sometimes, both at level of a single bee or the colony. Its negative symptoms are usually violent and visible in the short-term perspective [23,33–35]. Mitchell et al. [36] also confirmed that the 5 ppb concentration of imidacloprid is at the lower residual range of bee exposure estimates in the field. The feeding pattern we applied was intended to simulate nectar hoarding. No commercial agriculture existed within 12 km of the location and our bees did not have access to any significant natural food resources at the time of the study on the location, either. After six weeks of applying that diet, a few dozens of nurse workerbees were captured from the centers of the combs with open brood in each hive within IM-0, IM-5, and IM-200. They were subsequently destined for further biochemical analyses (3 hives x 3 feeding groups = 9 hive samples). The 6-week feeding period guaranteed that each sampled nurse workerbee received imidacloprid throughout its whole life from the moment the egg from which it emerged had been laid. On the other hand foragers are exposed to numerous external factors, hence the analyzing of the nurse worker bees was justified. It is also worth noting that during the six week study, we did not observe any marked differences in the syrup and pollen consumption between IM-0, IM-5, and IM—200. No visible losses of bees were noticed in the experimental colonies, either.

### The sample processing

Ten nurse workerbees showing proper activity and mobility were selected from each of the nine hive samples (30 workerbees from each feeding group x 3 groups = 90 workers) to collect hemolymph and obtain preparations of fat bodies.

To acquire fresh hemolymph, a glass capillary (20 µL; the 'end to end' type; without anticoagulant; Medlab Products, Raszyn, Poland) was inserted between the third and fourth tergite of a living worker, according to Łoś and Strachecka's [37] method. The hemolymph thus obtained from each single bee was immediately placed into a single sterile Eppendorf tube containing 25 µL of ice-cooled 0.6% NaCl, and then stored at −40 °C for further biochemical analyses after prior measurement of the sample volume. Subsequently, the same worker bees had their fat bodies dissected from the tergites between the third and fifth abdominal segments under a Stereo Zoom Microscope. The fat body tissue acquired from each workerbee was placed in a single sterile Eppendorf tube with 25 µl of ice-cooled 0.6% NaCl and subsequently homogenized at 4 °C. The homogenates were centrifuged for 1 min at 3000 g and the supernatants were frozen at − 40 °C at once, for further biochemical assays.

## Biochemical analyses

The concentrations of acetyl-CoA, IDH-2, AKG, succinate, fumarate, NADH2 and ATP, as well as the activities of COX and UQCR were evaluated in the hemolymph and fat-body supernatant samples obtained from the workerbees collected from the experimental colonies following the manufacturer's instructions for each specific kit (S1 Table).

## Statistical analyses

The Shapiro–Wilk test was applied to check the data distribution. Subsequently, one-way ANOVA followed by the LSD *post-hoc* test ($\alpha = 0.05$) was performed for each biochemical compound, considering feeding *group x tissue* as the experimental factors, i.e.: hemolymph IM-0, hemolymph IM-5, hemolymph IM-200, fat body IM-0, fat body IM-5, and fat body IM-200. The data base size for each of the nine compounds was: 180 items (30 items within each of the groups), df = 174.

Principal component analysis (PCA) was also performed to determine the relationships between the tissue types and measured compounds ($p = 0.05$) as well as their impact on the variability of the system described by PC1 and PC2. The PCA data matrix had 10 columns (parameters, i.e. the assayed compounds) and 185 rows (tissue types). The input matrix has been rescaled automatically. The optimal number of the main PCA components (i.e., PC1, PC2…..PCn) obtained in the analysis was determined based on the Cattel criterion. The Statistica software, version 12.0, StatSoft Inc., Tulsa, OK, USA, was used.

## Results

The values of the compounds involved in energy-metabolism were higher in the fat body than in the hemolymph of the workerbees sampled from our experimental colonies, whereas the compound variations were markedly higher in the hemolymph. This was particularly visible in the IM-0 group, in which the workers were not exposed to imidacloprid (Table 1; see also S2 Table).

The pattern of response to imidacloprid was the same in the hemolymph and the fat body irrespective of the diet type (Figs 1 and 2). Consequently, the *tissue x group* interactions did not occur (compare the interaction lines above the bars nested within a given compound and tissue at S1 and S2 Figs). On the other hand, the pattern of the response to imidacloprid was different and even opposite in IM-200 and IM-5 irrespective of the tissue type, as all the compounds were suppressed in IM-200, while in IM-5, the values of six of them were elevated and only three of them were lower. The response magnitude, in turn, was greater in the hemolymph than in the fat body. For more detailed information refer to S1 Fig, S2

**Table 1. Concentrations (c) or activities (a) of the biochemical compounds involved in the cellular respiration and energy metabolism and their variability in the workerbees sampled from the control groups, unexposed to imidacloprid.**

| | Acetyl-CoA(c) [nmol/mg] | | IDH-2(c) [ng/ml] | | AKG(c) [ng/ml] | | succinate(c) [µmol/l] | | fumarate(c) [µmol/l] | |
|---|---|---|---|---|---|---|---|---|---|---|
| | $\bar{x}$ | *W%* | $\bar{x}$ | *W%* | $\bar{x}$ | *W%* | $\bar{x}$ | *W%* | $\bar{x}$ | *W%* |
| HE | 133.4# | 7.8 | 11.1# | 14.6 | 8.2# | 9.0 | 8.06# | 7.4 | 4.29& | 15.6 |
| FB | 157.8# | 5.3 | 17.3# | 8.1 | 12.2# | 6.3 | 9.38# | 7.2 | 6.26& | 10.6 |
| | NADH2(c) [µmol/l] | | COX(a) [U/mg] | | UQRC(a) [U/mg] | | ATP(c) [nmol/mg] | | | |
| | $\bar{x}$ | *W%* | $\bar{x}$ | *W%* | $\bar{x}$ | *W%* | $\bar{x}$ | *W%* | | |
| HE | 14.4& | 7.2 | 1.44# | 11.7 | 0.82# | 10.7 | 6.3# | 10.6 | | |
| FB | 18.4& | 5.5 | 1.46# | 9.7 | 1.23# | 6.9 | 8.3# | 9.0 | | |

**Explanations:** HE – hemolymph. FB – fat body. Differences between the tissue means are significant for $p < 0.00001$ (#) or $p < 0.001$ (&). $\square$– means. W% – the variability coefficient (standard deviation expressed as the percentage of the appropriate mean's value). The abbreviations of the compound names are as follows: Acetyl-CoA – Acetyl Coenzyme A. Isocitrate dehydrogenase – IDH-2. Alpha-Ketoglutarate – AKG. Nicotinamides adenine dinucleotides – NADH2. Cytochrome C Oxidase – COX. Cytochrome C reductase – UQR. Adenosine triphosphate – ATP.

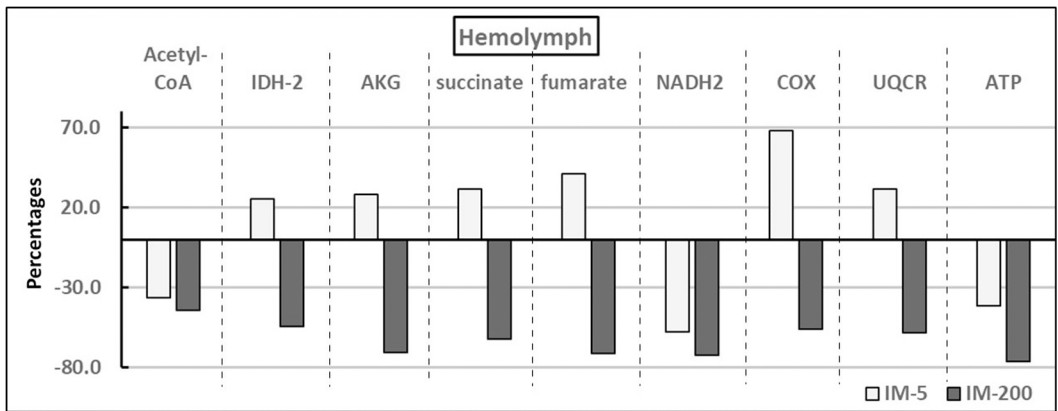

**Fig 1. The effect of exposure to imidacloprid on the hemolymph of workerbees sampled from colonies fed diets with different additions of imidacloprid. Explanations**: Each of the effects was shown as a difference between the mean in a given experimental group (IM-5 or IM-200) and the appropriate control group expressed as the percentage of the mean for this control group ($\frac{experimental\ group\ MEAN - control\ group\ MEAN}{control\ group\ MEAN} \times 100\%$). Each of these differences was significant at $p < 0.001$ – if they were not, they could not be considered here. The differences between IM-5 and IM-200 were also significant at $p < 0.05$ when compared within each of the compound separately. Detailed information about the statistical characteristics, including variability levels, is available in S1, S2 Figs and S2 Table, as well as in Table 1. The abbreviations used are as follows: Acetyl Coenzyme A (Acetyl-CoA). Isocitrate dehydrogenase (IDH-2). Alpha-Ketoglutarate (AKG). Nicotinamide adenine dinucleotide (NADH2). Cytochrome C Oxidase (COX). Cytochrome C reductase (UQRC). Adenosine triphosphate (ATP). The activities were evaluated for UQRC and COX, whereas it was the concentrations that were considered for the remaining biochemical compounds. The group in which the bees were fed with the diet containing 5 ppb of imidacloprid (IM-5). The group in which the bees were fed with the diet containing 200 ppb of imidacloprid (IM-200).

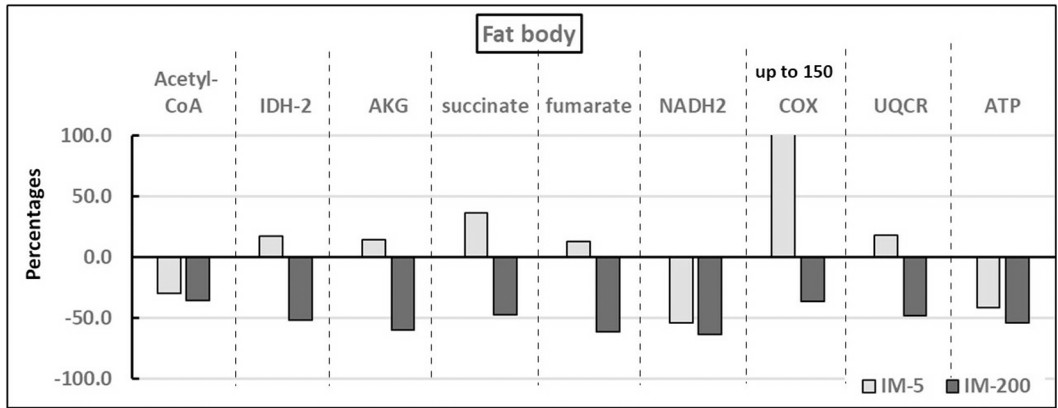

**Fig 2. The effect of exposure to imidacloprid on the fat body of workerbees sampled from colonies fed diets with different additions of imidacloprid. Explanations:** Each of the effects was shown as the difference between the mean in a given experimental group (IM-5 or IM-200) and the appropriate control group expressed as the percentage of the mean for this control group ($\frac{experimental\ group\ MEAN - control\ group\ MEAN}{control\ group\ MEAN} \times 100\%$).Each of these differences was significant at $p < 0.001$ – if they were not, they could not be considered here. The differences between IM-5 and IM-200 were also significant at $p < 0.05$ when compared within each of the compound separately. Detailed information about the statistical characteristics, including variability, is available in S1, S2 Figs and S2 Table, as well as in Table 1. The abbreviations used are as follows: Acetyl Coenzyme A (Acetyl-CoA). Isocitrate dehydrogenase (IDH-2). Alpha-Ketoglutarate (AKG). Nicotinamide adenine dinucleotide (NADH2). Cytochrome C Oxidase (COX). Cytochrome C reductase (UQRC). Adenosine triphosphate (ATP). The activities were evaluated for CoC and COX, whereas it was the concentrations that were considered for the remaining biochemical compounds. The group in which the bees were fed with the diet containing 5 ppb of imidacloprid (IM-5). The group in which the bees were fed with the diet containing 200 ppb of imidacloprid (IM-200).

[Fig](), and [S2 Table](). The most important of these observations is that the final energetic product, ATP, was suppressed in both tissues and both in IM-5 and IM-200. Finally, it is also worth emphasizing that the variability in the values of all the compounds was remarkably low in relation to their means; independently of the group both in the hemolymph and the fat bodies (see W% in [Table 1]() and [S2 Table]()).

PCA estimated nine components (PC1, PC2 … PC9) that explain 100% of the variability of our results, but importantly, the two main components PC1 and PC2 explain as much as 95% of this variability ([Figs 3]() and [4]()). Thus, the lack of influence of other uncontrolled factors on our results was indirectly confirmed, as summarized values of PC3 to PC9 described only 5% of variability. The analysis of PC1 and PC2 was used to determine the relationship between the type of diet and the studied chemical compounds. All the chemical compounds are plotted very near the circle that defines the area of maximum influence of every compound on PC1 and PC2 (the circle crossing |1| of PC1 and PC2), so all of them were very strongly affected by the diet types ([Fig 3]()). NADH2, ATP and acetyl-CoA are strongly and positively correlated (plotted very close to each other) and they are the only compounds located in the quadrant of negative values of PC1 and positive values of PC2. Again, most importantly, they were the only ones that were suppressed both in IM-5 and IM-200. On the other hand, COX, IDH-2, AKG, fumarate and succinate constituted the second group of strongly correlated compounds but they were placed in the quarter of the negative values of PC1 and PC2. Notably, it was only them, in turn, that were elevated in IM-5 but decreased in IM-200 (compare [Figs 1]() and [2]()). The results of PCA plotted in [Fig 4]() clearly confirmed that the type of the feeding group (IM-0, IM-5, and IM-200; PC1) had a much greater influence on the compounds affecting the cellular respiration and the energy metabolism than the tissue type (hemolymph and fat body). The influence of the

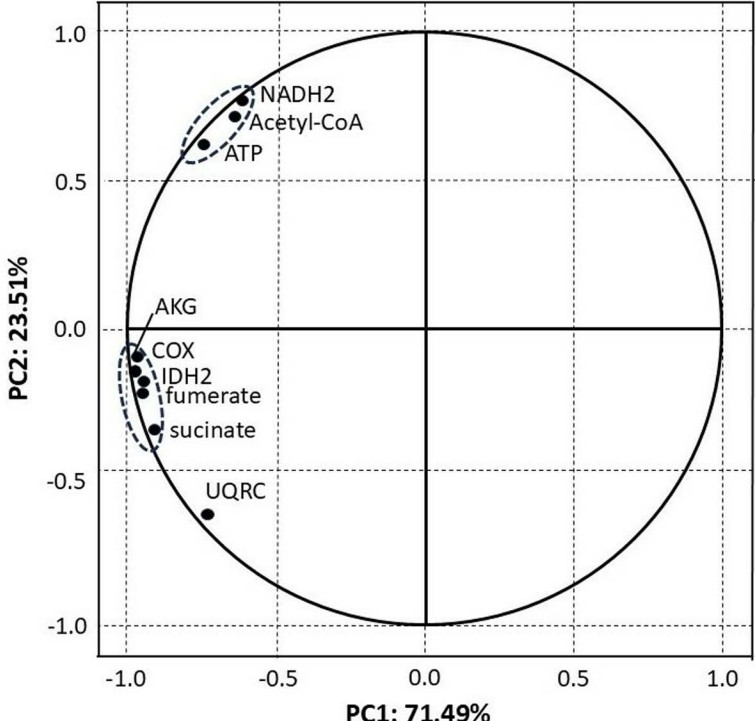

**Fig 3. PCA.** Projection of variables (biochemical compounds in our workerbees) on the PC1 and PC2 loading plot. **Explanations:** Acetyl Coenzyme A (Acetyl-CoA). Isocitrate dehydrogenase (IDH-2). Alpha-Ketoglutarate (AKG). Nicotinamide adenine dinucleotide (NADH2). Cytochrome C Oxidase (COX). Cytochrome C reductase (UQRC). Adenosine triphosphate (ATP). The activities were evaluated for UQRC and COX, whereas it was the concentrations that were considered for the remaining biochemical compounds.

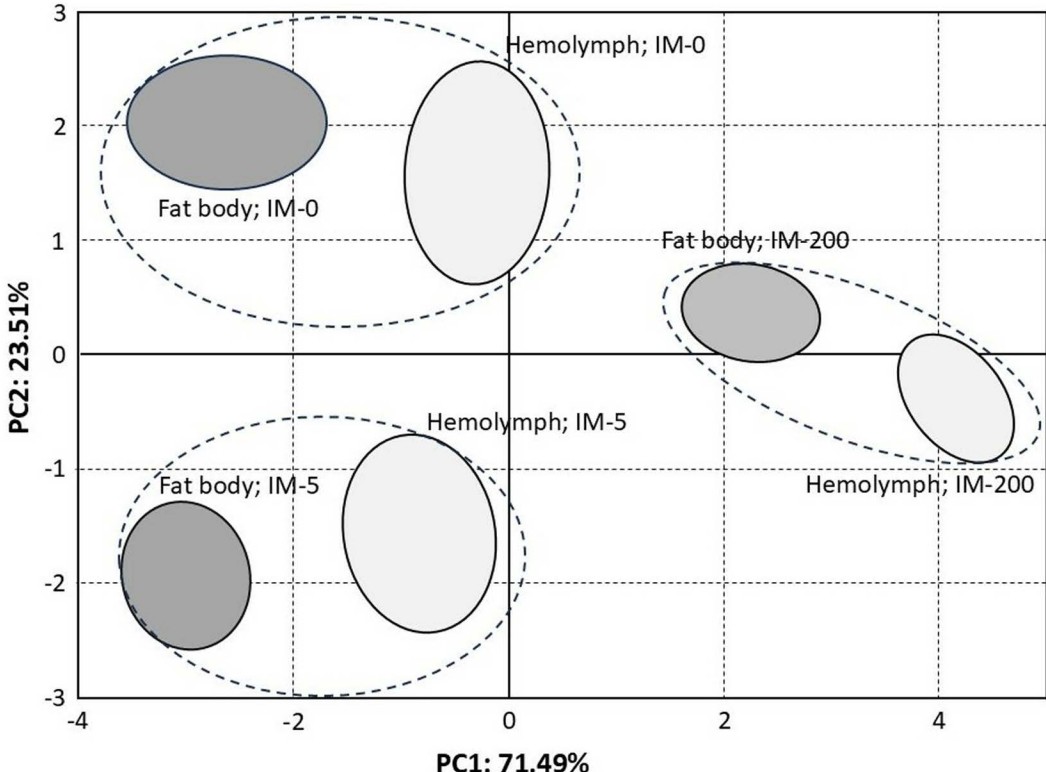

**Fig 4. Projection of the diet and tissue types on the PC1 and PC2 score plots. Explanations:** The group in which the bees were not given imidacloprid in their diet (IM-0). The group in which the bees were fed with the diet containing 5 ppb of imidacloprid (IM-5). The group in which the bees were fed with the diet containing 200 ppb of imidacloprid (IM-200).

feeding group, particularly IM-200, was really significant; both tissues within each feeding group were placed at the same regions of the plot, while the groups were located at different plot regions independently of the tissue. Consequently, a lack of *tissue x group* interactions was additionally confirmed again.

## Discussion

### The rationale for the field experiment

Laboratory cage tests are applied in honeybee research to make it possible to control environmental factors. However, the intracolony environment may affect many traits, particularly functional ones. Consequently, results obtained in field experiments may differ from those obtained in cages, even if individual, biochemical traits of a single worker are considered [38]. For instance, the cage environment had a destabilizing effect on the natural protease inhibitor system and decreased the worker resistance to microorganisms, which was not the case with the hive environment [39]. Hence, results of cage-based analyses of non-specific apian resistance should be treated with caution when used in reference to hive bees. Cresswell [40] also pointed out the problems about the relevance of laboratory-based results in toxicological research, and consequently considered "possibilities of providing a bridge between laboratory bioassay testing and full field experimentation". Therefore, in this research, we decided to study respiratory and citric cycle compounds in workerbees exposed to imidacloprid in field assays as we had already done in our previous studies [19,41], assuming that honeybees dwell in colonies kept in natural conditions rather and not in laboratory cages. We hoped that the results obtained in this way would be more applicable for the practical apiculture.

## Accuracy of our data

Comparing three honeybee colonies, Christen et al. [20] revealed different responses of the transcripts of genes involved in energy metabolism to sublethal doses of neonicotinoids in each of them. They also pointed out that individual bees consumed different amounts of the pesticide due to trophallaxis (compare Brodschneider et al. [42]) although they were all fed with the same diet with sublethal pesticide concentrations. Hence, the question might arise about the general value of our data which we obtained by analyzing individual workers from the three colonies within each feeding group (IM-0; IM-5; IM-200). First of all, all our colonies were similar both genetically and structurally. Secondly, the variability in all the compounds assayed in this study seemed very low in relation to their means, independently of the group and the tissue. Furthermore, PCA revealed that PC1 and PC2 explained as much as 95% of the total variability; each of the compounds influencing it very strongly (the component values were plotted very near the circle of the value │1.0│ of PC1 and PC2), the projection of the tissue x group spots on the main components' score plot was markedly scattered, their areas did not overlap, and notably they were plotted in different plot quarters. Consequently, despite Christen et al.'s [20] suggestions, our data can be considered accurate as no marked influences of uncontrolled factors occurred. Finally, it is worth mentioning that the bees in our previous studies [19,41] were fed with the same diets during a similar period of time and the imidacloprid content in the syrup was approximately 4.2 ppb and 196 ppb in IM-5 and IM-200, respectively, during the 3 months after syrup preparation. At the same time, the imidacloprid content was 4.1 ppb in IM-5 and 111.7 in IM-200 in the comb stores and it attained 0.35 ng/bee in the bee corpses, but only in IM-200. Therefore, it can be assumed that our bees were facing imidacloprid during entire experiment period, not only through the applied diets but also via the comb stores.

## Can we consider a more systemic response to imidacloprid?

As regards the present study, the patterns of the imidacloprid-dependent changes in the compounds involved in the cellular respiration and energy metabolism were the same in the hemolymph and in the fat bodies, which may indicate a systemic, and potentially fat-body-dependent response to imidacloprid of the entire honeybee organism or at least of many apian tissues. We have also shown that the values of these compounds were higher in the fat bodies than in the hemolymph independently of the feeding group, which may confirm this point of view. The honeybee fat body is notably the equivalent of the mammal liver, pancreas and adipose tissue and is responsible for the bee organism defense, particularly for the immune response, detoxification, antioxidation, and proteolysis [32]. Therefore, the high energy demand and production in this multifunctional tissue seems to be necessary for the maintenance of a high health status by bees (compare Arrese and Soulages [43]). Consequently, our study has shown that it may be particularly disadvantageous when bees are exposed to even residual doses of neonicotinoids in adverse environments. What is noteworthy is that the bee energy abilities are also related to high concentrations of proteins, glycogen and triglycerides in the fat body, especially in the third tergite [44]. The fat-body accumulation of these compounds is associated with the increase in the length and width of the trophocytes, as well as with the increase in the diameters of the oenocyte cell nuclei as a feature informing about the metabolic activities of these cells. Hence we suppose that neonicotinoids could also impair all these processes.

## Fat-body energy production *versus* worker fitness

Only acetyl-CoA, NADH2, and ATP were suppressed both in IM-5 and IM-200 in this study. It is symptomatic that PCA showed very high, mutual correlations just between these three compounds and it was only them that had positive PC2 and negative PC1 values. Insects produce energy from aerobic processes through oxidative phosphorylation [45]. Glucose obtained from trehalose is converted into pyruvate which is converted into acetyl-CoA that is oxidized to $CO_2$, $H_2O$, and finally ATP. Hence it is ATP that is crucial here while acetyl-CoA in combination with NADH2 are believed to be the pivotal components to obtain exactly ATP [45]. Due to the importance of NADH2 for energy production, its inhibition has been even used for the production of antidiabetic medicines [46]. Furthermore, its level was increased by adenosine

treatment reversing the harmful effects of imidacloprid on motion traits in honeybees [47]. Taking into account the above considerations, the bad news resulting from this study for contemporary honeybees is that imidacloprid not only decreased the values of all the nine compounds in IM-200, but also three of them which are those of the greatest importance for the energy supply in IM-5. In this way, we confirmed that the energy shortages caused by neonicotinoids applied even in sublethal doses, particularly those in the fat body tissue, could be one more biochemical mechanism which may worsen the honeybee bee health status, accelerating the depopulation of bee colonies in adverse environments (compare also with the drosophila study by Videlier et al. [17]). A good example of such a situation is *Nosema ceranae* and its specific adaptation to parasitic, intracellular cohabitation in honeybees. Instead of mitochondria, this microsporidian has the so-called mitosomes and is deprived of the oxidative phosphorylation pathway, and therefore ATP production, as well [48]. Consequently, its energy is acquired from the honeybee ATP through the parasite spores forming during development [49]. This results in risky, energy-consuming foraging, leading to higher forager mortality due to energetic stress followed by increasing hunger. Thus, our findings expand the existing knowledge, showing that imidacloprid may not only impair honeybee immunity [10] but also amplify the energy losses already caused by *N. ceranae*. Li et al. [27] and Christen et al. [24,25] confirmed that imidacloprid altered the expression of the oxidative phosphorylation transcripts. These findings are in agreement with the results of Fent et al. [7] showing that imidacloprid may contribute to the malfunction of mitochondria. Our studies point out for the first time that the malfunction could concern apian fat bodies.

### Imidacloprid-dependent hormesis don't concerns apian ATP

Although it is well documented knowledge that imidacloprid may harm honeybees by disrupting their physiology in every possible way [9,10,16,27,47,50], it is not entirely clear in which way sublethal doses of this insecticide influence bee metabolism, behavioral traits, fitness, and productivity.

On the one hand, IDH-2, AKG, succinate, fumarate, UQRC, and COX, i.e. six of the nine assayed compounds, were suppressed in IM-200, but their values increased significantly in IM-5 in this study. In their previous studies, Paleolog et al. [19,41] observed a similar, biphasic (stimulating at low doses but suppressing at high ones -hormesis) pattern of the response to imidacloprid of many proteolytic and antioxidative compounds, when the bees were fed with diets containing respectively 5 ppb or 200 ppb, of this insecticide. Kim et al. [30] have shown that changes in forager body weight, as well as those in the relevant target transcripts were also biphasic, but such a response pattern may not be favorable to colony health. A low, sublethal dose of clothianidin may stimulate learning, the motor function, and increase the flight efficiency in butterflies [29]. Honeybee colonies fed with a syrup containing 5 ppb of imidacloprid increased brood production and workerbees' activity, which did not result in increased food stores and colony survival, and more generally in colony productivity [23,28,51].

On the other hand, however, sublethal doses of neonicotinoids reduced honeybee foraging capacity and chances to successfully complete homing flights [5], which does not correspond with the butterfly results mentioned above. Low doses of imidacloprid down-regulated genes that modulate molting and energy metabolism in honeybee larvae and delayed their development [27], as well as strongly reducing the standard metabolic rate (the minimal energy expenditure required for the proper body-maintenance) in virgin and inseminated queen bees [26]. The metabolic rate is considered the key trait corresponding with fitness [17] and is important for shaping the life strategies of eusocial insects by evolution [18,19]. Imidacloprid also impaired the energy metabolism of bumblebees, and consequently, deteriorated the central carbon metabolism. The effect was stronger after the bumblebees were starved, i.e., their energy had to be allocated [34], probably just to improve self-maintenance. Martelli et al. [31] confirmed that imidacloprid applied even in sublethal doses may decrease the energy reserves by bonding to the brain receptors that may impair the nerve system functions in fruit flies.

Our study brings a new element to this discussion. Using our approach, in which, unlike many authors (compare Chen et al. [52]), we did not analyze the relevant transcripts of the brain genes but the key compounds of the

respiratory and citric cycle in the hemolymph and fat bodies, we revealed that imidacloprid may decrease the total energy production (particularly ATP) in honeybees although, in sublethal doses, it increased the levels of six of these nine compounds. So, examining the ATP level is pivotal here. These findings stay in line with the results of Powner et al. [53] who showed that bumblebees exposed to neonicotinoids were suffering from undermined mitochondrial function and reduced ATP production. Sargent et al. [54] stated that the efficiency of task performance, which markedly upscaled the energy demand, was dependent on efficient ATP supply, in the first place. Notably, Lin et al. [47] ameliorated the flight abilities and ATP (adenosine triphosphate) content previously deteriorated by imidacloprid by applying diet supplementation with adenosine in honeybees. Importantly, the same patterns of the imidacloprid-dependent changes in the compounds involved in the cellular respiration and energy metabolism observed in hemolymph and fat bodies may indicate a systemic, fat-body-dependent response to imidacloprid of the entire honeybee organism.

Hormesis, or the pharmacological effect, is a biphasic, physiological response to increasing concentrations of a compound or environmental stressor, which is usually stimulatory at low doses and inhibitory or harmful at higher ones [55]. Hormesis connected with the energy metabolism was suggested in previous studies of honeybees [51]. However, our study has shown that hormesis concerned the apian energy supplies, did not appear, particularly in the fat body tissue of worker bees exposed to the sublethal doses of imidacloprid. This was because the entire final energy production, i.e. production of ATP was suppressed, even though hormesis was observed in the case of many individual compounds involved in this process. For instance, the increase in COX was huge, particularly in the fat body tissue, but in spite of this, the total energy supply (ATP) significantly decreased in this study. What is worth noting is that a very high energy supply is necessary not only to fight pathogens and the proper body maintenance [10,18,28] but also to perform efficient flights. This is because, the energy needs are 80- to 100-fold higher when the flight muscles are working [45] and nicotinoid-related energy shortages leave too little of it available for flying [20,55]. Sargent et al. [54] found that imidacloprid exposure increases oxygen consumption in the brains and flight muscles of bumblebees, which explains their decreasing flight abilities, as well. It is the next bad information for beekeepers, who keep bees in an environment where nicotinoids occur that in this study imidacloprid, even in the sublethal, residual doses, led to deficits in the energy supply both in hemolymph and in fat body tissues, i.e. no hormesis occurred. This finding expands the knowledge of mechanisms of the pesticide harmfulness. This statement is in line with the findings of Lin et al. [47] who observed inadequate energy provision to the flight muscles of honeybees facing the imidacloprid stress, as the energy supply was relocated from motion to organism defense, i.e. to a component of body-maintenance. Finally, it should be mentioned that imidacloprid in doses that did not inhibit the flight ability of *Apis cerana* deteriorated the ability when it was accompanied with other pesticides [56], as well as different periods of exposure to imidacloprid applied in different studies mentioned above might possibly also influencing their outcomes.

## Conclusions

1. Imidacloprid may impair of the fat body energy metabolism and its energy supplies. In this way we have revealed one more mechanism, by which neonicotinoids may deteriorate functioning of this tissue, and consequently, the body-maintenance of honeybees when facing harmful, anthropogenic environments. This point of view can be confirmed by the same imidacloprid-dependent changes observed in hemolymph.

2. We showed that hormesis took place in most of the compounds of the respiratory and citric cycle when our bees were exposed to residual, sublethal doses of imidacloprid. The most important fact was, however, that despite this hormesis the sublethal doses of the neonicotinoid always decreased ATP production. Consequently, by interfering with energy supplies, they can harm both the honeybee health and body maintenance.

3. The approach we have proposed here, with the key compounds of the respiratory and citric cycle assayed, seems to be a good tool for better understanding mechanisms by which low, sometimes residual doses of pesticides may impair energy production in honeybees. More compounds ought to be included in further studies.

4. When examining how any xenobiotic influences any compounds involved in the energy metabolism, even when assaying gene transcripts, we recommend evaluating ATP levels at the same time, as it seems that ATP does not show hormesis even though the compounds involved in the respiratory and citric cycle can exhibit it. Therefore, this may be also a cost-effective method that can quickly yield results, particularly in the apiculture.

## Supporting information

**S1File. Imidacloprid impairs honeybee energy; text, S1 and_S2_Tables, S1 and S2_Figs.**
(PDF)

## Author contributions

**Conceptualization:** Jerzy Paleolog, Jerzy Wilde, Aneta Strachecka.

**Data curation:** Marek Gancarz.

**Formal analysis:** Jerzy Paleolog, Marek Gancarz.

**Funding acquisition:** Aneta Strachecka.

**Investigation:** Jerzy Paleolog, Jerzy Wilde, Aneta Strachecka.

**Methodology:** Jerzy Wilde, Aneta Strachecka.

**Project administration:** Jerzy Wilde, Aneta Strachecka.

**Resources:** Jerzy Wilde.

**Software:** Marek Gancarz.

**Supervision:** Jerzy Paleolog.

**Validation:** Aneta Strachecka.

**Visualization:** Jerzy Paleolog, Marek Gancarz.

**Writing – original draft:** Jerzy Paleolog.

**Writing – review & editing:** Jerzy Paleolog.

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
