## [Decision Letter · Decision Letter 0]

Dear Dr. Paleolog,

Thank you for submitting your manuscript to PLOS ONE. After careful consideration, we feel that it has merit but does not fully meet PLOS ONE’s publication criteria as it currently stands. Therefore, we invite you to submit a revised version of the manuscript that addresses the points raised during the review process.

We look forward to receiving your revised manuscript.

Kind regards,

Yahya Al Naggar

Academic Editor

PLOS ONE

“AS; The founding no. is: LKE.SUBB.WLE.22.058 by Ministry of National Education of the Republic of Poland, via University of Life Science in Lublin, Poland”

Reviewers' comments:

Reviewer's Responses to Questions

**Comments to the Author**

1. Is the manuscript technically sound, and do the data support the conclusions?

Reviewer #1: Yes

Reviewer #2: Yes

2. Has the statistical analysis been performed appropriately and rigorously?

Reviewer #1: Yes

Reviewer #2: Yes

3. Have the authors made all data underlying the findings in their manuscript fully available?

Reviewer #1: Yes

Reviewer #2: Yes

4. Is the manuscript presented in an intelligible fashion and written in standard English?

Reviewer #1: Yes

Reviewer #2: Yes

Reviewer #1: Reviewer Recommendation and Comments for Manuscript Number PONE-D-25-08140

Imidacloprid decreases the total energy production in western honeybees even though, in sublethal doses, it increased the values of six of the nine compounds in the respiratory and citric cycle

Please see the attachment.

Reviewer #2: This is an interesting article on a topical and important subject, leading to increased understanding of the effects of imidacloprid on the honey bee by using a novel approach studying the levels of various respiratory and citric cycle compounds in hemolymph and fat body tissue. It is mostly clearly presented and readable.

In some places abbreviations are inconsistent and some of the figures need corrections to the labelling.

The text is mostly clear. I have marked any typos that I noticed, and have also suggested some rewording in places, including a few places where the meaning of the text is not clear to me and needs reworded to make the meaning clear for the reader. Some of the text needs to be corrected.

Main changes requested are:

Abstract

L28: ID has not been defined.

L32-33: the notation changes here and needs made consistent with the rest and the main text.

L113: reword (see the text).

L116: can you add some references here, possibly some already cited in the text?

Methods

L127: indicate the timing of the experiment.

L181: where do the numbers 10 and 185 come from? They seem inconsistent with the 9 compounds and 180 samples mentioned earlier.

Results

L187: I suggest mentioning the control groups here, for clarity- see the text.

L192: Note that in Fig S1 and S2 legends, the letters above the bars need explained.

In Fig S1, some of the vertical axes say "Concetration" not "Concentration". Also in the legend "the pots" should say "the plots".

L201-202: reword for clarity.

L205-206: you refer to “compounds located in the quarter of negative values in PC1”. In fact from Fig 3 they are in the quadrant of negative values of PC1 and positive values of PC2- but the y-axis is mislabelled as PC1 and should say PC2. Also see comment at line 311.

L207: you refer to NDPH and IMH-2. It seems that IMH-2 should be IDH-2, and it is unclear what NDPH is - do you mean COX or CoC? There is no NDPH in Fig 3.

Also note that in Fig 1 and 2, you use the abbreviation CcO and in Fig 3 you use CoC. This is confusing.

L209: you refer to “the quarter of the positive values in PC1” but from Figure 3 they are in the quadrant of negative values for PC1 and negative values for PC2.

L234 and 247: As what follows is a series of notes, rather than sentences, maybe say here "The abbreviations used are as follows:" (and do the same for the Figure 2 legend) – see the text.

L236 and 249: you refer to CoC. The plot actually says CcO.

L256 and 262: note that Fig 3 vertical axis is mislabelled as PC1 and should be PC2.

Discussion

L279-280: unclear what “about 1.0” means here.

L284-289: after this can you say what you conclude from this?

L394-400: this sentence is long and unclear. Can you break it up and reword it?

L408: can you reword here? See text.

See other small points and suggestions marked on the text.

References

See a few small minor issues marked on the article.

Figure 1 and 2: need to change CcO to CoC.

Figure 3 and 4: need to correct vertical axis labelling.

For these and other, minor, points including rewording, please see the annotated article.

Supplementary figures

As noted above: in Fig S1 and S2 legends, the letters above the bars need explained.

In Fig S1, some of the vertical axes say "Concetration" not "Concentration". Also in the legend "the pots" should say "the plots".

**Do you want your identity to be public for this peer review?** For information about this choice, including consent withdrawal, please see our Privacy Policy

Reviewer #1: No

Reviewer #2: No

---

## [Author Response · Author response to Decision Letter 1]

15 Apr 2025

Response to Reviewers and Academic Editor – general remarks

PONE-D-25-08140

Imidacloprid decreases the total energy production in western honeybees even though, in sublethal doses, it increased the values of six of the nine compounds in the respiratory and citric cycle

PLOS ONE

1. Response to the Editor general comments:

After careful consideration, we feel that it has merit but does not fully meet PLOS ONE’s publication criteria as it currently stands…….

Authors: The manuscript has been reedited to match to the “Submission Guidelines”. All points raised during the review process have been duly considered – the manuscript has been completed and reworded. Additional references have been added as well.

Authors: The following statement has been added to the financial disclosure: "The funders had no role in study design, data collection and analysis, decision to publish, or preparation of the manuscript."

If applicable, we recommend that you deposit your laboratory protocols in protocols.io to enhance the reproducibility of your results. Protocols.io assigns your protocol its own identifier (DOI) so that it can be cited independently in the future. ….. Additionally, PLOS ONE offers an option for publishing peer-reviewed Lab Protocol articles, which describe protocols hosted on protocols.io..

Authors:

a) The analytical protocols of the kits producers were employed in this study. The appropriate links to the producers’ websites are given in the S1_Table within the Supporting Information.

b) The field and sampling procedures have already been published by us and the appropriate papers are cited in the manuscript – respectively [19], [41] and [37]

1. Please ensure that your manuscript meets PLOS ONE's style requirements,….

Authors: Manuscript has been duly reedited to meet PLOS ONE's style requirements

2. Thank you for stating the following financial disclosure: If this statement is not correct you must amend it as needed. Please include this amended Role of Funder statement in your cover letter; we will change the online submission form on your behalf.

Authors: the financial disclosure has been corrected as follows:

“AS; The founding no. is: LKE.SUBB.WLE.22.058 by Ministry of National Education of the Republic of Poland, via University of Life Science in Lublin, Poland. The funders had no role in study design, data collection and analysis, decision to publish, or preparation of the manuscript.”

The proper information is included within the Cover Letter

Authors: We have confirmed that our data are available at https://repod.icm.edu.pl/dataset.xhtml?persistentId=doi:10.18150/MMKUSE.

However the following DOI number have been activated during the review process: https://doi.org/10.18150/MMKUSE. Therefore, the better is to use it. We have informed this within the “Cover letter – revision”.

Authors: The proper captions have been added in the form of the Level 2 headings

2. Response to the Reviewer #2 included within the “Annotated article for the attention of the authors” (made within a .pdf file of the manuscript)

Abstracts:

Reviewer: Line 29 and 34 – Authors: the symbols have been corrected and defined

Introduction:

Reviewer: Line 114 and 119 – Authors: wording has been corrected exactly as the reviewer suggested

Reviewer: Line 116 –

Authors: one reference [32] that is a review paper concerning the knowledge about the latest researches on the apian fat body has been added.

Reviewer: Line 119 – Authors: wording has been corrected exactly as the reviewer suggested

Materials and methods:

Reviewer: Line 128 –

Authors: the proper items of information about this when the experiment took place (year, months, periods) was added in three places of M&M section.

Reviewer: Line 149 – Authors: wording has been changed exactly as the reviewer suggested

Reviewer: Line 182 –

Authors: PCA it is not the same like ANOVA. They complement each other. The values given in ANOVA and PCA do not have to be the same. So, information given by us is proper. PCA does not take into account the experimental variables and factors of the database of a given experiment - as the ANOVA does. Instead of this, PCA expresses the variability of the database on a scale from -1 to +1 and examines which, hypothetical, factors influence such variability in the greatest extent creating a series of hypothetical variance components ranking them from the most important to the least important one (PC1, PC2, ..... PCn; principal component analysis). Usually, we consider 2 (two-dimensional graph) or 3 (three-dimensional graph) of them being of the most importance; In this experiment we have considered two principal components (PC1 and PC2). Using this background, PCA analyses impact of the real experimental variables/factors. The further they are plotted from the value 0, the greater their influence is (share in the principal components). The closer they are plotted each other, the higher are the correlations between them, and so on. ……….The appropriate explanation has been added in the text and the text was a little bit reedited for better understanding.

Results and discussion:

Reviewer: Line 188 –

Authors: This not occurred in the control groups as the Reviewer suggested. It is misunderstanding. To clarify this we have changed the wording in the lines 188-190. The current version is: “The patterns of response to imidacloprid was the same in hemolymph and the fat body irrespective of the diet type. On the other hand, the response patterns were different and even opposite in IM-200 and IM-5 irrespective of the tissue type ……”. It should be clear for a reader now.

Reviewer: Line 194-195 –

Authors: Supporting Information - S1 Fig and S2 Fig.

Meaning of the letters above the bars were explained within the Figs’ legends. In S1 Fig, "Concetration" has been replaced by "Concentration" (vertical axes). Compound nomenclature has been adjusted to match that applied in the revised version of the manuscript. The meaning of the lines above the bars (S1 Fig and S2 Fig) has been explained. The expression “(compare the interaction plots at S1 Fig and S2”) has been reworded to (“compare the interaction lines above the bars nested within a given compound and tissue at S1 and S2 Figs)”. Some part of the manuscript has been moved up. The proper explanation has been added.

These changes stay also in line with remarks of Reviewer#2 as well.

Reviewer: Line 202-204:

Authors: The sentence has been reworded as follows: “Furthermore, all the chemical compounds are plotted very near the circle that defines the area of maximum influence (it crossing │1│of PC1 and PC2), so all of them were very strongly affected by the diet types.”

Reviewer: Line 206-210 –

Authors: Nomenclature and description of this quadrant of this graph has been corrected. The manuscript has been corrected.

Reviewer: Line 234-328 –

Authors: This part of manuscript has been reworded according the Reviewer suggestions. Nomenclature of the compounds was corrected (also within the Tables).

Reviewer: Line 249: Authors: Nomenclature was corrected

Reviewer: Line 256, 262: Authors: All figures have been corrected where necessary.

Reviewer: Line 281:

Authors: This has been reworded as follows: the component values were plotted very near the circle of the value │1.0│ of PC1 and PC2

Reviewer: Line 282: Authors: This has been corrected according the Reviewer suggestion

Reviewer: Line 289:

Authors: the following has been added; “Therefore, it can be assumed that our bees were facing imidacloprid during entire experiment period, not only by the applied diets but also but the comb storages.”

Reviewer: Line 300: Authors: This has been corrected according the Reviewer suggestion

Reviewer: Line 311:

Authors: The sentence after correction is as follows: “that had positive PC2 and negative PC1 values.”

Reviewer: Line 316: Authors: The typing error has been corrected

Reviewer: Line 223:

Authors: The sentence has been reworded according the Reviewer suggestion as follows: “three of them which are those of the greatest importance....”

Reviewer: Line 340: Authors: The typing error has been corrected

Reviewer: Line 366: Authors: the word “metabolic” has been added as Reviewer has suggested

Reviewer: Line 374: Authors: The typing error has been corrected

Reviewer: Lines 392 to 401: Authors: this part was reedited and reworded

Reviewer: Lines 408- 409

Authors: The sentence after rewording is: “It is the next bad information news for beekeepers, who keep bees dwelling in an environment where nicotinoids occur that….

Reviewer: Lines 412: Authors: The typing error has been corrected.

Reviewer: Line 419: Authors: The expression after rewording is: "possibly also influencing"

Reviewer: Line 422 Authors: word “influences” has been added

Reviewer: - References ……

Authors: references were completely reedited

2. Response to the Reviewer #1 (made within the WORD review file)

Reviewer: why did the authors feed the colonies sugar syrup with imidacloprid rather than by rearing caged bees? I ask this question because the author does not present any data at the colony level in the article?

Authors:

1.The information about the colonies’ strength and a syrup consumption are given in the manuscript.

2. The following paragraph has been added to the discussion:

“Laboratory cage tests are applied in honeybee research to make it possible to control environment factors. However, the intra colony environment may affect many traits, particularly functional ones. Consequently, results obtained in field experiments may differ from those obtained in cages, even if individual, biochemical traits of a single worker are considered [38]. For instance, he cage environment had a destabilizing effect on the natural protease inhibitor system and decreased the worker resistance to microorganisms, which was not the case with the hive environment [39]. Hence, results of cage-based analyses of non-specific apian resistance should be treated with caution when used in reference to hive bees. Cresswell [40] also pointed out the problems about the relevance of laboratory-based results in toxicological research, and consequently considered “possibilities of providing a bridge between laboratory bioassay testing and full field experimentation”. Therefore, in this research, we decided to study respiratory and citric cycle compounds in workerbees exposed to imidacloprid in field assays, as we already had done in our previous studies [19, 41], assuming that honeybees dwell in colonies kept in the natural conditions rather and not in laboratory cages. We hoped that the results obtained in this way would be more applicable for the practical apiculture.”

The added references:

[38] Dziechciarz P., Borsuk G., Olszewski K. (2019). Prospects and validity of laboratory cage tests conducted in honeybee research part one: main directions of use of laboratory cage tests in honeybee research. J. Apic. Sci. 63, (2), 201-207.]

[39] Strachecka A. Paleolog J., Borsuk G., Olszewski K. Grzywnowicz K, Gryzińska M. (2011) Body-surface protease inhibitors in cage and hive Apis mellifera L. Acta Sci. Pol., Zootechnica 10 (4), 125–132

[40] Cresswell, J.E. A meta-analysis of experiments testing the effects of a neonicotinoid insecticide (imidacloprid) on honey bees. Ecotoxicology 20, 149–157 (2011). https://doi.org/10.1007/s10646-010-0566-0

Reviewer: The authors mentioned in lines 145-148 that they captured nurse worker bees for laboratory analysis, but there are no analysis results concerning nurse worker bees sampled from the experimental colonies were presented in this section. Why is that ?

Authors: It is some kind of misunderstanding. The only results concerning nurse workerbees have been shown in the “Results” section. To solve the problem we changed the following:

a). The aim (Introduction) has been changed and its current wording is: “The aim of this study was to compare the concentrations/activities of the above compounds in workerbees captured from the colonies fed with a diet containing 200 ppb, 5 ppb or 0 ppb of imidacloprid. Both the workerbee hemolymph and fat bodies were assayed.”

b). Paragraph 2.3. of M&M has been changes and is wording is as follows: “The concentrations of acetyl-CoA, IDH-2, AKG, succinate, fumarate, NADH2 and ATP, as well as the activities of COX and UQCR were evaluated in the hemolymph and fat-body supernatant of workerbees sampled from the experimental colonies following the producer instructions for each specific kit (S1 Table).”

c) Finally, the first part of the results section has been reedited and its wording is: “The values of compounds involved in energy-metabolism were higher in the fat body than in the hemolymph of workerbees sampled from our experimental colonies, whereas the compound variations were markedly higher in the hemolymph. This was particularly seen in the IM-0 group, in which the workers were not exposed to imidacloprid (Table 1).

d) Consequently the table and figures titles have been changed in agreement with the pints a, b, and c.

e) The expression “laboratory analysis” has bee replaced by the expression “biochemical analysis”, which seems to be more adequate. The expression “laboratory analysis” may suggest the cage test

Reviewer: Why was the exposure duration to imidacloprid set at 6 weeks instead of 5 weeks or 7 weeks?

Authors: The following explanation has been added in M&M (the line 148): “The 6-week feeding period guaranteed that each sampled nurse workerbee received imidacloprid throughout its whole life from the moment the egg, she emerged, had been laid. On the other hand foragers are exposed to numerous external factors, hence analyzing of the nurse worker bees was justified.”

Reviewer: What was the strength of the experimental colonies established? How many combs does each hive contain?

Authors: The text beginning from the line 126 has been reedited as follows: “The synthetic colonies of similar strength, as well as worker and brood structure, were set up in northeastern Poland (19.53 E, 53.50 N) in June 2024. Each of these colonies fully populated a one-box hive with ten 360 mm x 260 mm frames and was headed by an egg laying one-year-old purebred queen. All the queens belonged to the same Apis mellifera carnica commercial stock and were all obtained from the same mother-queen. In the first ten days of June, after removing the hive food-stores, the colonies were given sugar/water syrup (5:3 w/w)”

Title

Reviewer: The original title of the authors' work is "Imidacloprid decreases the total energy production in western honeybees even though, in sublethal doses, it increased the values of six of the nine compounds in the respiratory and citric cycle." This title does not align with the actual content of the study. The authors state in the ti

---

## [Decision Letter · Decision Letter 1]

Dear Dr. Paleolog,

Thank you for submitting your manuscript to PLOS ONE. After careful consideration, we feel that it has merit but does not fully meet PLOS ONE’s publication criteria as it currently stands. Therefore, we invite you to submit a revised version of the manuscript that addresses the points raised during the review process.

We look forward to receiving your revised manuscript.

Kind regards,

Yahya Al Naggar

Academic Editor

PLOS ONE

Journal Requirements:

Reviewers' comments:

Reviewer's Responses to Questions

**Comments to the Author**

Reviewer #1: All comments have been addressed

Reviewer #3: (No Response)

2. Is the manuscript technically sound, and do the data support the conclusions?

Reviewer #1: Yes

Reviewer #3: Yes

3. Has the statistical analysis been performed appropriately and rigorously?

Reviewer #1: Yes

Reviewer #3: Yes

4. Have the authors made all data underlying the findings in their manuscript fully available?

Reviewer #1: Yes

Reviewer #3: Yes

5. Is the manuscript presented in an intelligible fashion and written in standard English?

Reviewer #1: Yes

Reviewer #3: Yes

Reviewer #1: The authors have revised the manuscript in great detail, and the quality of the article has been greatly improved.Therefore, I suggest that the editor may now publish the paper in PLOS ONE.

Reviewer #3: General Comments:

The premise of the study seems solid, and the methods and analysis seem fine, from my knowledge of this area. The authors have correctly identified the importance of this study - adding to existing information on an important topic (imadacloprid effects on bees), while also providing a valuable example on the more general principle of hormesis.

Further, the suggestions of the original reviewers appear to have been addressed - either the manuscript was changed as requested (usually), or the authors responded with justification for their approach. On that basis, the manuscript meets the requirements outlined by the original reviewers.

I believe the main area of weakness is in some of the communication - the writing is awkward in many places. Some corrections were pointed out by the original reviewers and corrected. However, there is still significant text that would benefit greatly from rewording. For instance, iL101-3 “Hence, more studies are necessary to answer whether the energy supply decreased by the sublethal doses of the pesticide or just hormesis may occur in this case?” is grammatically difficult to understand. There are many instances like this. It was not possible as a reviewer to suggest corrections for all of these, I suggest engaging a writing editor who can dedicate some time to making those corrections.

Specific comments

L30-31. Example of awkward wording here. Stating “The goal of the study…” rather than just bringing up a question, would help in the comprehension of the study.

L70-76. This can be shortened, the first two sentences for instance are very general statements that we should assume the reader knows already.

L84. The paragraph starting here is very long, and in fact is addressing two separate questions or points. The first has to do with conflicting results or responses when low doses of IM are used, and the second is how the response is measured (transcriptosome, in most cases). Split up these paragraphs. In the first paragraph, be clear the purpose in mentioning these various studies - that there are variable results that point towards a complicated response to IM.

L150 There are two terms used here that are slightly different - sublethal and adverse - not mutually exclusive. Is the 5ppb adverse in some way, even though sublethal? And is the 200 ppb concentration also sublethal? It would be clearer to refine that terminology.

L227. This is, I Assume, subtext for the table, and the paragraph after this then continues the main text of the results.

Fig 1,2 . In both of these, it seems like showing some measure of variability and statistical significance on the graph is needed. As I interpret it currently, it only shows the mean difference (as a ratio), and does not indicate on the graph which are statistically significant.

L297. This would seem to be better placed in the introduction of the article. At the least, it would read better to make the main justification (which is valid) for using field experiments earlier, and then more shortly reiterate it in the discussion.

L315. This paragraph could be shifted later. If it is necessary to defend the accuracy of the data in this way, that reads more smoothly if mentioned later, after the main result of the study is discussed (which starts in the next paragraph, L342)

L448. Leave “Adaptive” off the list here. It’s not clear that hormesis is definitely found to be adaptive in all instances where it is studied. It could be a nonadaptive byproduct of other processes.

L477. It seems like the “biphasic” response is a major and obvious finding of the study, but is not mentioned in the conclusions.

**Do you want your identity to be public for this peer review?** For information about this choice, including consent withdrawal, please see our Privacy Policy

Reviewer #1: No

Reviewer #3: No

---

## [Author Response · Author response to Decision Letter 2]

28 May 2025

PONE-D-25-08140R1

Imidacloprid decreases the energy production in hemolymph and fat body of western honeybees even though, in sublethal doses, it increased the values of six of the nine compounds in the respiratory and citric cycle

PLOS ONE

Authors:

A small correction of the title was made by the Editor as follows:

“Imidacloprid decreases the energy production in the hemolymph and fat body of western honeybees even though, in sublethal doses, it increased the values of six of the nine compounds in the respiratory and citric cycle

Journal Requirements:

Authors: The reference list (chapter “References”) in the first submission of our manuscript contained 53 references. The reference list submitted in our revised manuscript (MAJOR REVISION) contained 56 references because the following 3 references were added in response to the doubts of Reviewer #2:

[38] Dziechciarz P., Borsuk G., Olszewski K. (2019). Prospects and validity of laboratory cage tests conducted in honeybee research part one: main directions of use of laboratory cage tests in honeybee research. J. Apic. Sci. 63, (2), 201-207.]

[39] Strachecka A. Paleolog J., Borsuk G., Olszewski K. Grzywnowicz K, Gryzińska M. (2011) Body-surface protease inhibitors in cage and hive Apis mellifera L. Acta Sci. Pol., Zootechnica 10 (4), 125–132

[40] Cresswell, J.E. A meta-analysis of experiments testing the effects of a neonicotinoid insecticide (imidacloprid) on honey bees. Ecotoxicology 20, 149–157 (2011). https://doi.org/10.1007/s10646-010-0566-0

The Reviewer #2 has accepted this changes.

No references were replaced or retracted, although the order of citing publications was changed a little due to corrections made in response to the Reviewers' #1 and #2 comments.

During the second MINOR REVISION no references were added, replaced or retracted, as well as the order of citing did not changed.

Reviewer #3: General Comments:

Rev. #3: The premise of the study seems solid, and the methods and analysis seem fine, from my knowledge of this area. The authors have correctly identified the importance of this study - adding to existing information on an important topic (imidacloprid effects on bees), while also providing a valuable example on the more general principle of hormesis.

Further, the suggestions of the original reviewers appear to have been addressed - either the manuscript was changed as requested (usually), or the authors responded with justification for their approach. On that basis, the manuscript meets the requirements outlined by the original reviewers.

Rev. #3: I believe the main area of weakness is in some of the communication - the writing is awkward in many places. Some corrections were pointed out by the original reviewers and corrected. However, there is still significant text that would benefit greatly from rewording.

For instance, iL101-3 “Hence, more studies are necessary to answer whether the energy supply decreased by the sublethal doses of the pesticide or just hormesis may occur in this case?” is grammatically difficult to understand.

Authors: The sentence has been reworded as follows:

“Consequently, more studies are necessary to answer whether body energy supply is decreased by sublethal doses of the pesticide or whether such doses can also increase the supply as a result of hormesis.”

Rev. #3: There are many instances like this. It was not possible as a reviewer to suggest corrections for all of these. I suggest engaging a writing editor who can dedicate some time to making those corrections.

Authors: The licensed editor has made corrections that are within a marked-up copy of our manuscript labeled 'Revised Manuscript with Track Changes'. The corrections of the editor are marked by “HP” (blue).

Reviewer #3: Specific comments

Rev. #3: L30-31. Example of awkward wording here. Stating “The goal of the study…” rather than just bringing up a question, would help in the comprehension of the study.

Authors: The sentence has been reworded as follows:

“Therefore, our goal was to answer which of these two phenomena occurs in the hemolymph/fat body and at what doses of imidacloprid”.

Rev. #3: L70-76. This can be shortened, the first two sentences for instance are very general statements that we should assume the reader knows already.

Authors: These sentences have been deleted.

Rev. #3: L84. The paragraph starting here is very long, and in fact is addressing two separate questions or points. The first has to do with conflicting results or responses when low doses of IM are used, and the second is how the response is measured (transcriptosome, in most cases). Split up these paragraphs. In the first paragraph, be clear the purpose in mentioning these various studies - that there are variable results that point towards a complicated response to IM.

Authors: This paragraph has been splinted into two different once. In the first one the following construction has been applied:

“There is, however, a problem with the exposition of honeybees to sublethal, residual, field-relevant doses of imidacloprid (about 5 ppb), as different results of the exposition were obtained in different studies. This implies a yet incompletely explored complex response to this pesticide. On the one hand, unlike the higher doses, which have consistently proven to be highly detrimental, they ………..”

Rev. #3: L150 There are two terms used here that are slightly different - sublethal and adverse - not mutually exclusive. Is the 5ppb adverse in some way, even though sublethal? And is the 200 ppb concentration also sublethal? It would be clearer to refine that terminology.

Authors: This part of manuscript has been reworded and developed as follows:

“This concentration sometimes turn out to be adverse for honeybee colonies either impairing the physiology of an individual bee or weakening entire colonies – particularly in a long-term perspective [15, 16, 28]. Notably, its negative symptoms may be hidden. On the other hand, the concentration of 200 ppb is considered harmful or severely harmful and even lethal sometimes, both at level of a single bee or the colony. Its negative symptoms are usually violent and visible in the short-term perspective [23,33–35].”

Rev. #3: L227. This is, I Assume, subtext for the table, and the paragraph after this then continues the main text of the results.

Authors: Yes, it is subtext for the table, in fact the legend (Explanations) for tis table.

Rationale: The “Results” chapter is short and consists of two parts. The first concentrate on general analysis, and the second one, on PCA results. Most of information is included within the Table and Figs (with their explanations), but not in the contents of the "Results" manuscript. Consequently, we have placed the Table after the first part, where it was cited in, but all Figs were placed after the second, last part, which in fact, constituting only a short paragraph. Please note that legends and captions for the Table and Figures take up almost as much space as the “Results” manuscript contents. Therefore, such an arrangement to allow a reader to read this short text without long brakes – “having it in almost “one piece”. Reviewer 1 and 2 haven’t addressed no comments to such an arrangement, so it seems that they accepted it. Of course the PLOS technical editor will make the final decision.

Rev. #3: Fig 1,2 . In both of these, it seems like showing some measure of variability and statistical significance on the graph is needed. As I interpret it currently, it only shows the mean difference (as a ratio), and does not indicate on the graph which are statistically significant.

Authors: We suggest not to show either measures of variability or the letters pointing the statistical significance within Fig 1,2. Instead of this we have completed the information given in “Explanations” for these Figs, as we agree with the Rev 3 that a reader should obtain some statistical information here. Now, this part of the “Explanations” stands as follows:

Each of these differences was significant at p < 0.001 – if they were not, they could not be considered here. The differences between IM-5 and IM-200 were also significant at p < 0.05 when compared within each of the compound separately. Detailed information about the statistical characteristics, including variability levels, is available in S1, S2 Figs and S2 Table, as well as in Table 1.”

Rationale: The values plotted at Figs 1, 2 aren’t original data obtained in our experiment, but are result of their transformation. We subtracted the control group mean from the mean of an adequate experimental group obtaining its deviations from the control. Consequently, we have used averages, but not the source data during the further calculations. This procedure was performed for each chemical compound separately, and therefore the results cannot be compared or plotted at one figure, as different compounds have completely different scales and measurement units. Therefore, next transformation was performed (data standardization) to express each of the above differences as its percentage for the mean of the adequate control group. In this way we obtained the transformed data that have no units and are expressed in the same scale (percentage). Now one was able to compare practically everything within one figure. However, as a result of such a transformation variability is changed, so plotting it, or performing statistical comparisons, do not make sense. Although a different procedure can be used enabling the statistical comparisons, but then, the next transformation (arcsin x) have to be performed. One way to another, we will have to analyze the double transformed data and the obtained information may be confusing.

Secondly, variability of the control groups is shaped by the external factors in different ways in the case of different chemical compounds. The procedure described above allow to better show (“extract”) the effect of the experimental factor (equalizing control to 0), as well as comparing all chemical compounds at the same plot. This help to avoid the information noise connected with analyzing of many plots (separate plot for every compound) focusing on the central message of the paper. However, a reader can find all necessary statical comparisons, information about variability, and information about interactions obtained during analyzing of the original, untransformed data at S1 Table and S1, S2 Figs included within the “Supporting Information”. Table 1 preceding Figs 1 and 2 within the main manuscript also constitute an auxiliary material in this context.

Rev. #3: L297. This would seem to be better placed in the introduction of the article. At the least, it would read better to make the main justification (which is valid) for using field experiments earlier, and then more shortly reiterate it in the discussion.

Authors: This paragraph, i.e. sub-chapter entitled “The rationale for the field experiment” (current version reworded by editor), was added as the first subchapter of “Discussion” according the following suggestion of Reviewer #1:

“why did the authors feed the colonies sugar syrup with imidacloprid rather than by rearing caged bees? I ask this question because the author does not present any data at the colony level in the article?”.

We have added this sub-chapter (paragraph) just right here, as it isn’t involved to the rationale of the study goal, or working hypotheses, or the asked questions. Although it contains important supplementary information, is not needed for the main flow of narration/reasoning within the "Introduction." Therefore, we would like to leave this information within the “Discussion”, as we want a reader to focus on the main information flow leading her/him to the rationale of study goal when reading the "Introduction”. However, leaving the main considerations within the “Discussion“ but taking into account the Rew. #3 comments, we have decided to add a short paragraph at the end of the “Introduction”. This paragraph is as follows:

“Many toxicological experiments on pesticides in honeybees were performed in cage or semi-cage experiments [14, 22] that allowed to better monitor/control the external conditions. However, results obtained in fully functional colonies in the field could be different. Therefore, we decided to perform a field experiment in this study, controlling environmental factors to the maximum extent.”

Rev. #3: L315. This paragraph could be shifted later. If it is necessary to defend the accuracy of the data in this way, that reads more smoothly if mentioned later, after the main result of the study is discussed (which starts in the next paragraph, L342)

Authors: We had been urged to discussing of accuracy of original field data by the reviewers of our previous papers, since the accuracy of such a data can be unsatisfactory. Therefore, we created the second subchapter of the “Discussion” entitled “Accuracy of our data” placed after the first one entitled “The rationale for the field experiment”, since it also contained important supplementary information.

Summing up, we would like to leave these two chapters containing supplementary, important information at the beginning of the “Discussion”, if possible.

Rationale: We believe that it is better familiarize the reader with the supplementary information at the beginning of the “Discission” and only then continue the main narration. In this case a reader contionuoing the Ddiscussion move consistently from issue to issue until finally meet conclusions and the central message of our work. This seems us to be a better way of the narration flow than that, when after discussing the main issues of the paper one make the break to familiarize a reader with the supplementary issues, and subsequently, come back to the main narration flow again, to drove the conclusions.

Rev. #3: L448. Leave “Adaptive” off the list here. It’s not clear that hormesis is definitely found to be adaptive in all instances where it is studied. It could be a nonadaptive byproduct of other processes.

Authors: This has been corrected

Rev. #3: L477. It seems like the “biphasic” response is a major and obvious finding of the study, but is not mentioned in the conclusions.

Authors: The conclusion 2 has been substantially developed to reflect this issue. Now it stands as follows:

“2. We showed that hormesis took place in most of the compounds of the respiratory and citric cycle when our bees were exposed to residual, sublethal doses of imidacloprid. The most important fact was, however, that despite this hormesis the sublethal doses of the neonicotinoid always decreased ATP production. Consequently, by interfering with energy supplies, they can harm both the honeybee health and body maintenance.”

[While revising your submission, please upload your figure files to the Preflight Analysis and Conversion Engine (PACE) digital diagnostic tool, https://pacev2.apexcovantage.com/. PACE helps ensure that figures meet PLOS requirements. To use PACE, you must first register as a user. Registration is free. Then, login and navigate to the UPLOAD tab, where you will find detailed instructions on how to use the tool. If you encounter any issues or have any questions when using PACE, please email PLOS at figures@plos.org. Please note that Supporting Information files do not need this step.

---

## [Editor Report · Decision Letter 2]

Imidacloprid decreases energy production in the hemolymph and fat body of western honeybees even though, in sublethal doses, it increased the values of six of the nine compounds in the respiratory and citric cycle

PONE-D-25-08140R2

Dear Dr. Paleolog,

We’re pleased to inform you that your manuscript has been judged scientifically suitable for publication and will be formally accepted for publication once it meets all outstanding technical requirements.

Within one week, you’ll receive an e-mail detailing the required amendments. When these have been addressed, you’ll receive a formal acceptance letter, and your manuscript will be scheduled for publication.

Kind regards,

Yahya Al Naggar

Academic Editor

PLOS ONE
---

## [Editor Report · Acceptance letter]

PONE-D-25-08140R2

PLOS ONE

Dear Dr. Paleolog,

I'm pleased to inform you that your manuscript has been deemed suitable for publication in PLOS ONE. Congratulations! Your manuscript is now being handed over to our production team.

Kind regards,

on behalf of

Dr. Yahya Al Naggar

Academic Editor

PLOS ONE